# dattri: A Library for Efficient Data Attribution

**Junwei Deng**[1*]  **Ting-Wei Li**[1*]  **Shiyuan Zhang**[1]  **Shixuan Liu**[2]  **Yijun Pan**[2]

**Hao Huang**[1]  **Xinhe Wang**[2]  **Pingbang Hu**[1]  **Xingjian Zhang**[2]  **Jiaqi W. Ma**[1]

[1]University of Illinois Urbana-Champaign    [2]University of Michigan
[*]Equal Contribution

## Abstract

Data attribution methods aim to quantify the influence of individual training samples on the prediction of artificial intelligence (AI) models. As training data plays an increasingly crucial role in the modern development of large-scale AI models, data attribution has found broad applications in improving AI performance and safety. However, despite a surge of new data attribution methods being developed recently, there lacks a comprehensive library that facilitates the development, benchmarking, and deployment of different data attribution methods. In this work, we introduce dattri, an open-source data attribution library that addresses the above needs. Specifically, dattri highlights three novel design features. Firstly, dattri proposes a unified and easy-to-use API, allowing users to integrate different data attribution methods into their PyTorch-based machine learning pipeline with a few lines of code changed. Secondly, dattri modularizes low-level utility functions that are commonly used in data attribution methods, such as Hessian-vector product, inverse-Hessian-vector product or random projection, making it easier for researchers to develop new data attribution methods. Thirdly, dattri provides a comprehensive benchmark framework with pre-trained models and ground truth annotations for a variety of benchmark settings, including generative AI settings. We have implemented a variety of state-of-the-art efficient data attribution methods that can be applied to large-scale neural network models, and will continuously update the library in the future. Using the developed dattri library, we are able to perform a comprehensive and fair benchmark analysis across a wide range of data attribution methods. The source code of dattri is available at https://github.com/TRAIS-Lab/dattri.

## 1   Introduction

*Data attribution* is a family of methods that aim to quantify the influence of individual training samples on the output of artificial intelligence (AI) models. As training data is becoming increasingly critical for the advancement of modern AI models, especially for large foundation models [19], there has been a surge of data attribution methods developed recently [22, 29, 36, 27]. These methods have found broad data-centric applications in improving the performance and safety of AI models, such as noisy label detection [22], data selection [9], and copyright compensation [6].

However, a comprehensive infrastructural library that facilitates the development, benchmarking, and deployment of efficient data attribution methods is lacking. This absence hinders the standardization and acceleration of research in this area, creating inefficiencies and inconsistencies in how data attribution methods are developed and applied. Although there have been a couple of prior efforts [33, 28, 18] for unifying APIs and benchmarking, many missing opportunities remain unaddressed,

motivating the need for this work. We defer to Section 2.2 for a detailed comparison between the existing libraries and our work.

In this work, we introduce `dattri`, an open-source data attribution library with the following design objectives. Firstly, for *users* deploying existing data attribution methods, we aim to provide a unified and user-friendly API across different methods. Specifically, our API design emphasizes **minimal code invasion**, i.e., allowing the data attribution methods to be integrated into most common PyTorch-based machine learning pipelines with only a few lines of code changed. This is non-trivial as data attribution methods often require access to internal information of the models, such as gradients or hidden representations. We achieve this goal by providing helper decorators to streamline the integration. Secondly, for *researchers* developing new data attribution methods, we aim to facilitate the development by providing efficient implementations of **low-level utility functions**, such as Hessian-vector product, inverse-Hessian-vector product or random projection. These functions are used by multiple existing data attribution methods and will likely be useful in the development of new methods. In `dattri`, we implement the data attribution methods in a carefully designed modular fashion, providing both a set of modularized low-level functions and examples of the usage of these functions. Finally, for both *users* and *researchers*, we aim to provide a benchmark suite that highlights **a comprehensive list of evaluation metrics** and diverse benchmark settings, including **generative AI settings**. In addition to the code for the evaluation metrics and benchmark experiments, we also provide the **trained model checkpoints** for each benchmark setting. Since some evaluation metrics for data attribution require hundreds or even thousands of model retraining, the provided trained model checkpoints could significantly reduce the computational burden of benchmark evaluation. The **bolded features** listed above are all novel designs in `dattri` compared to existing literature.

We have implemented a variety of data attribution methods, evaluation metrics, and benchmark settings in `dattri`. Our library currently covers four families of data attribution methods, each named after a representative method within its category. These families are: Influence Function (IF) [22], TracIn [29], Representer Point Selection (RPS) [36], and TRAK [27]. We have excluded certain other methods, notably the game-theoretic methods such as Data Shapley [10, 17] or Data Banzhaf [34], to focus on computationally efficient methods that do not require extensive model retraining. We have implemented three evaluation metrics commonly used in the data attribution literature: noisy label detection [22], leave-one-out (LOO) correlation [22], and linear datamodeling score (LDS) [15]. We provide six benchmark settings on different combinations of models and datasets, with a variety of machine learning tasks, including image classification, text generation, and music generation. With the developed `dattri` library, we have performed a comprehensive and fair benchmark analysis across the methods and settings. Our results suggest that IF performs well on linear models, while TRAK generally outperforms other methods in most experimental settings.

In summary, `dattri` is a comprehensive library with numerous novel features tailored to facilitate the development, benchmarking, and deployment of efficient data attribution methods. We will also continuously update this library to include more efficient data attribution methods and benchmark settings in future iterations.

## 2 Related Work

In this section, we briefly review data attribution methods in Section 2.1 and compare `dattri` with existing data attribution libraries in Section 2.2.

### 2.1 Data attribution methods

**The data attribution problem.** Suppose we have a training set $\mathcal{S} = \{x_1, \ldots, x_n\}$, a test set $\mathcal{T} = \{x_1, \ldots, x_m\}$, and a trained model output function $f_\Theta$ that is parameterized by $\Theta$. Typically, a data attribution method $\tau$ derives the attribution scores $\tau(x, \mathcal{S}; f_\Theta) \in \mathbb{R}^n$, where $x \in \mathcal{T}$, to quantify the influence of each training data point in $\mathcal{S}$ on the model output on $x$.

**Data attribution methods.** Our library aims to cover a diverse set of representative data attribution methods while recognizing that it is impossible to implement all existing methods. Notably, we have omitted one popular family of game-theoretic methods, including Data Shapley [10, 16] and Data Banzhaf [34]. These methods often require repeatedly removing subsets of training data and

Table 1: A summary of the efficient data attribution methods available in `dattri`. These methods are clustered into four families: IF, TracIn, RPS, and TRAK. The empirical experiments are demonstrated separately by different evaluation metrics and models. The experimental settings are stated in Section 3.3, and the results are detailed in Section 4. Here, we use the symbols "-/+/++" to qualitatively indicate the performance of each efficient data attribution method to be "similar to random/better than random/much better than random". The "Linear" column is based on the result of logistic regression (LR), while the "Non-linear" column is based on that of a variety of neural network models.

| Family | Algorithms | LOO | | LDS | | AUC | |
|---|---|---|---|---|---|---|---|
| | | Linear | Non-linear | Linear | Non-linear | Linear | Non-linear |
| IF | Explicit [22] | ++ | - | ++ | - | ++ | - |
| | CG [26] | ++ | - | ++ | + | ++ | + |
| | LiSSA [1] | ++ | - | ++ | + | ++ | + |
| | Arnoldi [30] | + | - | + | - | ++ | + |
| TracIn | TracInCP [29] | + | - | + | + | ++ | + |
| | Grad-Dot [5] | + | - | + | + | ++ | + |
| | Grad-Cos [5] | + | - | + | + | - | - |
| RPS | RPS-L2 [36] | + | - | + | - | ++ | + |
| TRAK | TRAK [27] | ++ | - | ++ | ++ | ++ | ++ |

retraining the model on the remaining data, making them computationally infeasible for large-scale applications.

Prioritizing efficient data attribution methods applicable to large neural network models, we focus on the following four families of methods. We have implemented most of the state-of-the-art methods.

One of the most popular and widely used data attribution families is the Influence Function (IF) [22], which approximates the influence by calculating the Hessian matrix and the gradient of data samples. Since explicitly calculating the Inverse-Hessian-Vector-Product can be prohibitively heavy in terms of computational load and memory usage, many alternative methods are proposed to alleviate the computation and memory cost. Some of the popular ones include Conjugate Gradients (CG) [26], LiSSA [1], Arnoldi [30]. Another family of data attribution methods, TracIn [29], assumes the hessian matrix to be an identity matrix and proposes to leverage multiple checkpoints during the training and assume the hessian matrix to be an identity matrix. Existing literature [5] also proposes two simplified versions, i.e., "Grad-Dot" and "Grad-Cos". "Grad-Dot" can be seen as TracIn with only one checkpoint, and "Grad-Cos" additionally normalizes the score with the gradient norm. Representer Point Selection (RPS) [36] is another family of data attribution methods. It uses the representer point theorem for kernels to represent the pre-activation prediction as a linear combination of training samples. TRAK [27] is the last family; it leverages the empirical neural tangent kernel approximation and random projection to improve efficiency and efficacy. The detailed formula definition of each data attribution method is stated in Appendix A.

In Table 1, we summarize these methods and provide a qualitative overview of their performance based on our benchmark experiments detailed in Section 4.

## 2.2 Data attribution libraries

There are three existing libraries aiming to benchmark or unify the implementations of different data attribution methods, as summarized in Table 2. Specifically, `OpenDataVal` [18] primarily focuses on game-theoretic methods. While it also implements a couple of IF variants, it misses most of the efficient data attribution methods. The scale of the benchmark settings of `OpenDataVal` are also mostly small. `pyDVL` [33] implements both the IF family of methods and game-theoretic methods, but it does not have a benchmark component. The methods implemented by `influenciae` [28] are closer to ours, yet we cover more data attribution methods as well as significantly more comprehensive benchmark datasets, tasks, and metrics (so far `influenciae` only has one benchmark and metric). Moreover, `influenciae` is based on Tensorflow and has only one evaluation metric for image classification, which limits its applicability and flexibility. In contrast, our library is built on PyTorch,

Table 2: A summary of existing libraries and our library, `dattri`. Our library covers a broader set of efficient data attribution methods and a more comprehensive benchmark suite.

| Library | Algorithm | | | | | Framework | Benchmark | |
| --- | --- | --- | --- | --- | --- | --- | --- | --- |
| | IF | TracIn | RPS | TRAK | Game-Theoretic | | Model Type | Metrics |
| `pyDVL` | Yes | / | / | / | Yes | PyTorch | / | / |
| `OpenDataVal` | Partial | / | / | / | Yes | PyTorch | Classification | Noisy Label/Feature Detection Point Removal/Addition |
| `Influenciae` | Yes | Yes | Yes | / | / | Tensorflow | Image Classification | Noisy Label Detection |
| `dattri` | Yes | Yes | Yes | Yes | / | PyTorch | Image Classification Text Generation Music Generation | Noisy Label Detection Linear Datamodeling Score (LDS) Leave-one-out (LOO) Correlation |

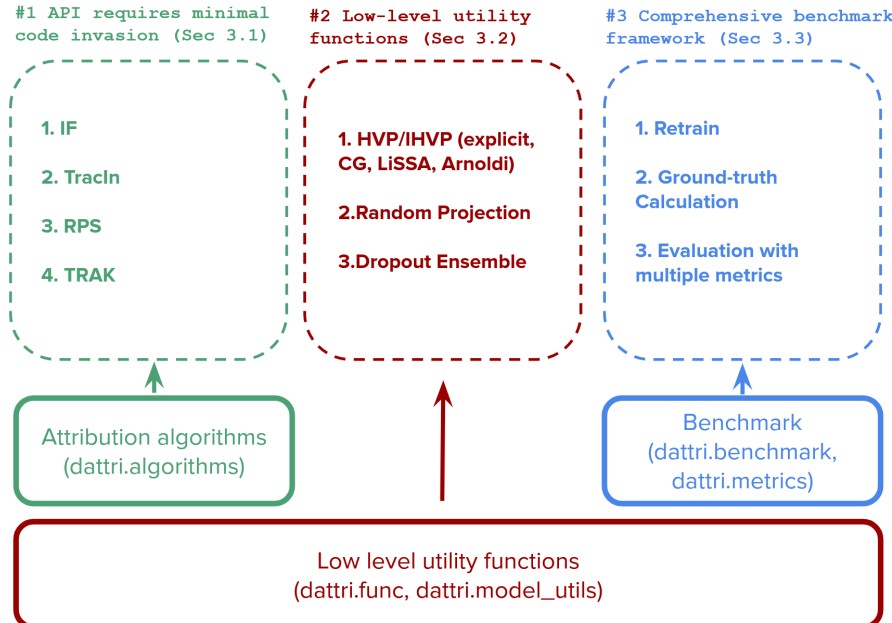

Figure 1: Architecture of `dattri` and the functionalities of each module in `dattri` serve.

and includes a rich family of efficient data attribution methods and a comprehensive benchmarking component. Our library also highlights novel design features as mentioned in Section 1 and detailed in Section 3.

In addition, there is a remotely relevant library, `Captum` [23], that primarily focuses on machine learning model interpretability. It implements several data attribution methods as part of its suite of interpretability tools, alongside other techniques such as feature and neuron attribution methods [11]. However, the goals and scope of `Captum` differ significantly from those of `dattri`.

## 3 Design of `dattri`

In this section, we introduce the design of `dattri` that provides a unified and user-friendly API (Section 3.1), modularized low-level utility functions (Section 3.2), and a comprehensive benchmark suite (Section 3.3).

### 3.1 A unified and user-friendly API

Data attribution methods often require internal gradients or hidden representations of the model to calculate the attribution scores. Consequently, many existing implementations of data attribution methods are heavily *invasive* to the model training pipeline, i.e., the data attribution process is significantly entangled with the model training code, making it challenging for users to adapt the code to other models or applications.

`dattri` is carefully designed to provide a unified API that can be applied to the most common PyTorch model training pipeline with minimal code invasion. Demo 1 shows an example of applying IF methods on a PyTorch model.

Specifically, the user will first define an `AttributionTask` object. This object contains necessary attribution task information such as the loss function from which the model is trained, the model architecture, and the trained model checkpoints. Next, the user will initialize an `Attributor` instance with the `AttributionTask` object and additional configuration parameters. Note that each `Attributor` class corresponds to a specific attribution method. Finally, the `Attributor` will perform data attribution using the training and test data loaders, which typically come directly from the model training pipeline.

The same API works for all the data attribution methods available in `dattri`, so that users can easily switch across different methods.

```python
def f(params, data): # an example of loss function using CE loss
    x, y = data
    loss = nn.CrossEntropyLoss()
    yhat = torch.func.functional_call(model, params, x)
    return loss(yhat, y)

attr_task = AttributionTask(
    loss_func=f,
    model=model,
    checkpoints=model.state_dict(),
    target_func=f # the target function to be attributed could differ from the loss
        function (e.g., this could be defined as the prediction logit)
)
attributor = IFAttributorCG(
        task=attr_task,
        device=torch.device("cuda"),
        **attributor_hyperparams # e.g., regularization, ...
) # similar for other attributors
attributor.cache(train_loader) # optional pre-processing to accelerate the attribution
score = attributor.attribute(train_loader, test_loader)
```

Demo 1: Example usage of `dattri` to perform attribution on a PyTorch model. Users will first define a `AttributionTask` object, `task`, that collects necessary configuration information about the attribution task. Next, users can initialize a specific attributor class (in this demo, `IFAttributorCG` corresponds to the influence function with CG method) that takes `task` as the input. Finally, users feed the training and testing data loaders (typically from the model training pipeline) to `attributor` and obtain the attribution scores.

## 3.2    Modularized low-level utility functions

Different data attribution methods can share common sub-routines in their algorithms. In `dattri`, we modularize such sub-routines through low-level utility functions so that they can be reused in the development of new methods.

There are two types of low-level utility functions, respectively, implemented in the modules `dattri.func` and `dattri.model_utils`. The `dattri.func` module is built on top of `torch.func`[1], which allows flexible mathematical manipulation of numerical functions. This is a helpful abstraction as data attribution methods often utilize mathematical operations beyond standard PyTorch operators (e.g., operations involving higher-order derivatives). We implement a few such mathematical operations in `dattri.func`, including Hessian-vector product (HVP), inverse-Hessian-vector product (IHVP) and random projection. The `dattri.model_utils` module, on the other hand, implements model-level manipulations that have been shown to be useful in recent literature. Below, we provide more details about several key low-level utility functions.

---

[1]See `https://pytorch.org/docs/stable/func.html` for more details about `torch.func`.

**HVP/IHVP.** Mathematically, given a target function $f_\Theta(x)$ with the Hessian denoted as $H(x;\Theta) = \nabla^2_\Theta f_\Theta(x)$, and a vector $v$, the HVP function is defined as $\text{HVP}(x, v; \Theta) = H(x;\Theta)v$; while the IHVP function is defined as $\text{IHVP}(x, v; \Theta) = H(x;\Theta)^{-1}v$. We implement the HVP function with a thin wrapper on the composition of Jacobian-vector product functions available in `torch.func`. We further implement a variety of efficient approximated algorithms for the IHVP functions (such as Conjugate Gradients (CG) [26] or LiSSA [1]), most of which re-use HVP as a sub-routine. For each of these IHVP algorithms, we implemented two versions under `dattri.func.hessian`, `ihvp_{alg_name}` and `ihvp_at_x_{alg_name}`. As can be seen in Demo 2 (with CG as an example), `ihvp_cg` takes the target function as input and returns a function $\text{IHVP}(x, v; \Theta)$ (i.e., `ihvp_func` in Demo 2). On the other hand, `ihvp_at_x_cg` further takes the data and parameters as input and returns a function `ihvp_at_x_func` that only takes $v$ as input. The latter implementation serves a specific need of data attribution methods where we want to calculate $\text{IHVP}(x, v; \Theta)$ for multiple $v$'s with the same $x$ and $\Theta$. This implementation allows us to pre-process and cache intermediate results that only depend on $x$ and $\Theta$ to accelerate the algorithm.

```python
from dattri.func.hessian import ihvp_cg, ihvp_at_x_cg

def f(x, param): # target function
    return torch.sin(x / param).sum()

x = torch.randn(2)
param = torch.randn(1)
v = torch.randn(5, 2)
# ihvp_cg method
ihvp_func = ihvp_cg(f, argnums=0, max_iter=2) # argnums=0 indicates that the param
    of (x, param) to be passed to ihvp_func is the model parameter
ihvp_result_1 = ihvp_func((x, param), v) # both (x, param) and v as the inputs
# ihvp_at_x_cg method: (x, param) is cached
ihvp_at_x_func = ihvp_at_x_cg(f, x, param, argnums=0, max_iter=2)
ihvp_result_2 = ihvp_at_x_func(v) # only v as the input
# the above two will give the same result
assert torch.allclose(ihvp_result_1, ihvp_result_2)
```

Demo 2: Example usage of the CG implementation of the IHVP function.

**Random projection.** Some data attribution methods, such as TRAK [27] and TracIn [29], involve inner product among gradients of model parameters. This calculation can be significantly accelerated by dimension reduction through random projection when the model parameter size is extremely large. We provide a simple wrapper on top of the random projection toolkit, `fast_jl`, implemented by Park et al. [27].

Researchers can leverage this utility function into the development of new data attribution methods when dealing with high-dimensional model parameters.

**Dropout ensemble.** Recent studies [31, 27] have shown that the efficacy of many data attribution methods can be significantly improved by ensembling multiple independently trained models with different random seeds. Furthermore, a recent paper [7] proposes *dropout ensemble*, which utilizes multiple dropout masks on the same model to perform ensembling, leading to superior efficiency-efficacy trade-off in comparison to naive ensembles. In `dattri.model_utils`, we provide a utility function `activate_dropout` to enable dropout ensemble for different data attribution methods.

### 3.3 A comprehensive benchmark suite

**Data attribution metrics.** As an emerging research area, data attribution has been evaluated by a variety of metrics in the literature. Among them, there are two types of mainstream evaluation metrics. The first type of metrics treats the change of model outputs after removing certain data points and retraining the model as a gold standard for quantifying the influence of individual training samples:

- Leave-one-out (**LOO**) correlation [22]: This metric refers to the Pearson correlation between the predicted model output difference (by data attribution method) and the model output with leave-one-out training.

- Linear datamodeling score (**LDS**) [15]: This metric aims at probing data attribution methods' ability to make counterfactual predictions based on the attribution score derived from the learned model output function $f_\Theta$ and the corresponding dataset to train $f_\Theta$. Because most data attribution methods are assumed to be *additive*[2], the data attribution scores can be used to predict the model output function learned from a subset of training data in a summation form. More details of LDS are deferred to Appendix B.

The second type of metrics evaluates data attribution methods through downstream applications, where the most common ones are noisy label detection and data selection [22, 10]. However, a recent study [35] demonstrates that the data selection task is problematic. Therefore we focus on noisy label detection only in our benchmark:

- Area under the ROC curve (**AUC**) for noisy label detection: This metric is specifically for noisy label detection tasks. For this task, a certain portion of the training samples' labels are flipped and data attribution methods are utilized to prioritize the training points with higher self-influence for humans to inspect when repairing the dataset. The task can thus be treated as a ranking problem (based on the magnitude of attribution scores) and evaluated by AUC.

**Diverse experimental settings.** `dattri` introduces diverse experimental settings including image classification, music generation, and text generation, which are listed in Table 3. To summarize, we consider a series of models with different architectures trained on diverse datasets: (1) a logistic regression (LR) classifier and a three-layer MLP classifier trained on the MNIST-10 dataset [25], (2) a ResNet-9 classifier [13] trained on CIFAR-10 [24] and CIFAR-2 dataset (a two-class subset of the CIFAR-10 dataset), (3) a Music Transformer [14] trained on the MAESTRO dataset [12] and (4) a NanoGPT [21] trained on the Shakespeare dataset [20]. In particular, the former two are supervised image classification settings, while the latter two are generative settings. For the classification settings, we sample 5000 training samples and 500 test samples from MNIST-10 and CIFAR-10/CIFAR-2 datasets. For the MAESTRO dataset, we sample 5000 training samples and 178 generated samples. For the Shakespeare dataset, we use the full training set with size 3921 and sample 435 generated samples. More detailed setups for each dataset and model are listed in Appendix C.

Table 3: The full experimental setting for data attribution benchmark.

| Dataset | Model | Task | Sample size (train,test) | Parameter size | Metrics | Data Source |
|---------|-------|------|--------------------------|----------------|---------|-------------|
| MNIST-10 | LR | Image Classification | (5000,500) | 7840 | LOO/LDS/AUC | [8] |
| MNIST-10 | MLP | Image Classification | (5000,500) | 0.11M | LOO/LDS/AUC | [8] |
| CIFAR-2 | ResNet-9 | Image Classification | (5000,500) | 4.83M | LDS | [24] |
| CIFAR-10 | ResNet-9 | Image Classification | (5000,500) | 4.83M | AUC | [24] |
| MAESTRO | Music Transformer | Music Generation | (5000,178) | 13.3M | LDS | [12] |
| Shakespeare | NanoGPT | Text Generation | (3921,435) | 10.7M | LDS | [20] |

**Pre-trained models with ground truth.** For the aforementioned benchmark settings, we provide pre-trained models, and ground truth annotations corresponding to each evaluation metric. For MNIST-10 experiments, we provide pre-trained models produced by exact leave-one-out training (i.e., 5000 models for each of the LR and MLP experiments). Across all settings, we pre-train 100 models for LDS calculation on a random half-dataset controlled by a fixed random seed generation procedure. The first 50 models are used in our benchmark experiments to assemble and compute the attribution scores. For example, TRAK and TracIn can utilize multiple models to improve their performance on the same task. The last 50 models are used for LDS calculation, which requires several models that are trained on a portion of the original training dataset. The formulation of LDS is detailed in Appendix B.

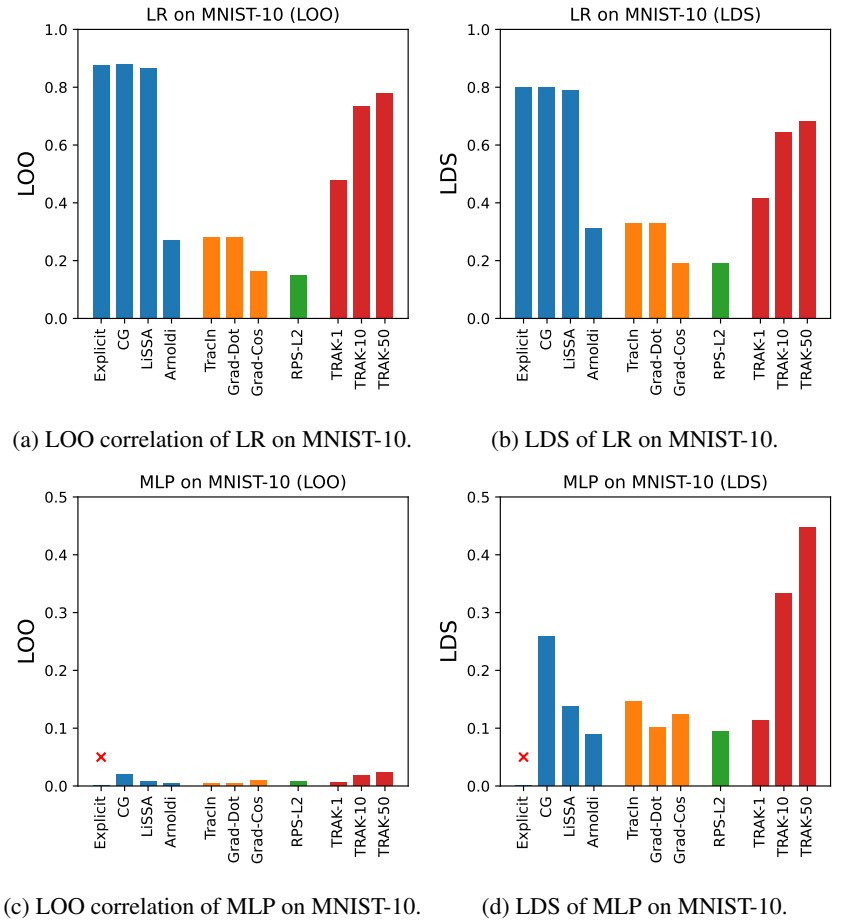

(a) LOO correlation of LR on MNIST-10.

(b) LDS of LR on MNIST-10.

(c) LOO correlation of MLP on MNIST-10.

(d) LDS of MLP on MNIST-10.

Figure 2: The LOO correlation and LDS evaluation of each efficient data attribution method on LR and MLP trained on MNIST-10. The red cross indicates that the experiment runs out of the time or memory budget.

# 4    Benchmark Experiments

## 4.1    Experimental setup

**Datasets and models.**    We follow the experimental settings listed in Table 3.

**Data attribution methods.**    We benchmark data attribution methods listed in Table 1. Some of the methods consist of a couple of hyperparameters related to their numerical stability. During the benchmarking, we mildly tune the hyperparameters to avoid falling into the numerically unstable region for each method. The details of the hyperparameter tuning are stated in Appendix C.1. Furthermore, some methods become infeasible in terms of computation time or memory as the model size and data size grow. In this case, their result is marked as a red cross in the plots. The TRAK method can trade off computation for efficacy by ensembling several independently trained models. We denote TRAK without ensembling as "TRAK-1" while TRAK with 10 or 50 model ensembling respectively as "TRAK-10" and "TRAK-50".

---

[2]If a data attribution method is additive, then it defines an attribution score that the overall influence of a group is the sum of the individual influence in the group.

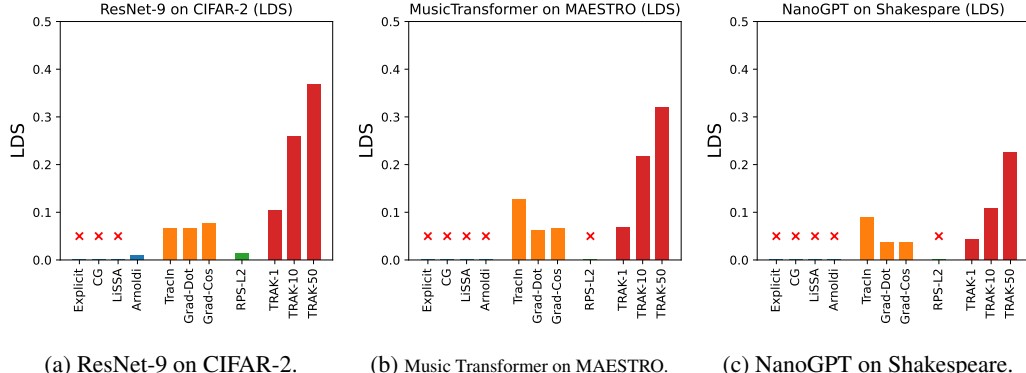

(a) ResNet-9 on CIFAR-2.  (b) Music Transformer on MAESTRO.  (c) NanoGPT on Shakespeare.

Figure 3: The LDS of each efficient data attribution method on ResNet-9 trained on Cifar-2, Music Transformer trained on MAESTRO, and NanoGPT trained on Shakespare. The red cross indicates that the experiment runs out of time or memory budget.

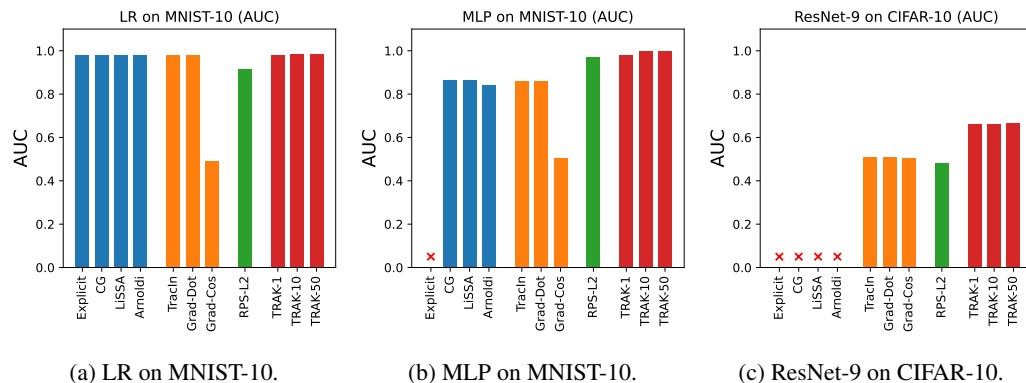

(a) LR on MNIST-10.  (b) MLP on MNIST-10.  (c) ResNet-9 on CIFAR-10.

Figure 4: The noisy label detection AUC evaluation of each efficient data attribution method on LR and MLP trained on MNIST-10 and ResNet-9 trained on CIFAR10. The red cross indicates that the experiment runs out of time or memory budget.

## 4.2 Experimental results

**LOO/LDS performance on logistic regression (LR) and MLP.** We first investigate the experimental setting of LR classifiers trained on MNIST-10, a linear model setting (Figure 2a and Figure 2b). All methods have better-than-random performance in terms of LOO and LDS, i.e., positive LOO and LDS. The IF and its variants (except for Arnoldi) perform better than other methods. TRAK-10 and TRAK-50 achieve comparable results to IF methods, while TRAK-1 is worse than IF.

The experiment on MLP, a non-linear model, on the same dataset shows different results (Figures 2c and 2d). None of the methods work well in terms of LOO. The LDS performance of most methods also drops significantly in comparison to the results on LR. TRAK-50 achieves the best performance in this setting.

These results align with the community understanding that IF methods are brittle for complicated non-convex models [3] and that the LOO metric is less informative for the non-convex regime [2].

**LDS performance on larger models.** We also evaluate data attribution methods in terms of LDS in larger experimental settings, including a larger image classification setting (ResNet-9 on CIFAR-2) and two generative settings (Music Transformer on MAESTRO and NanoGPT on Shakespeare). As shown in Figure 3, IF and its variants are not feasible for these settings because of the computational cost. Among all the efficient data attribution methods implemented in `dattri`, only TRAK can achieve non-trivial LDS results with the help of ensembling.

**AUC performance.** We further evaluate the data attribution methods in terms of the AUC in the downstream noisy label detection task, as shown in Figure 4. Except for "Grad-Cos"[3], most methods have better-than-random performance ($> 0.5$) for the experiments on LR/MLP on MNIST-10, with some methods achieving near-perfect AUC (close to 1). For the experiments on ResNet-9 on Cifar-10, there is a significant performance drop for all methods, with only TRAK achieving non-trivial performance ($> 0.5$).

Overall, we found that the IF family performs well in small experimental settings and linear models, while TRAK generally outperforms other methods in most experimental settings.

Additionally, there are some limitations to the current evaluation metrics. Both the LOO and LDS metrics require a large number of retrained models to obtain the "ground truth." The AUC metric, meanwhile, is tied to a specific downstream application and is only applicable to classification tasks. To address these limitations, and as a key contribution of `dattri`, we provide pre-trained model checkpoints associated with the LOO and LDS metrics, allowing users to bypass the costly retraining process for these evaluations.

## 5 Conclusion

In this work, we introduce `dattri`, a comprehensive open-source library that facilitates the research, development, and deployment of data attribution methods. The main contribution of `dattri` is three-fold: (1) a unified and user-friendly API for seamless integration into PyTorch-based ML pipelines, (2) modularized implementations of low-level utility functions to aid researchers in developing new methods, and (3) a fair benchmark suite with diverse evaluation metrics, experimental settings, and pre-trained model checkpoints. `dattri` addresses critical infrastructural needs in the data attribution domain, offering a collaborative platform that promotes standardization and accelerates the development/deployment of data attribution methods.

**Limitations and future work.** While `dattri` has implemented a rich family of existing data attribution methods and experimental settings, admittedly there are still a number of efficient data attribution methods and benchmark datasets missing in our current library. As for future work, we will continuously incorporate new methods, benchmarking experiments, and evaluation metrics, advancing the state-of-the-art of efficient data attribution and unlocking the potential for large-scale data-centric AI applications.

**Acknowledgement**

The authors would like to thank Juhan Bae and Kristian Georgiev for their helpful discussions.

This work was in part supported by NCSA Delta GPU at NCSA through allocation CIS230197 from the Advanced Cyberinfrastructure Coordination Ecosystem: Services & Support (ACCESS) program [4], which is supported by National Science Foundation grants #2138259, #2138286, #2138307, #2137603, and #2138296.

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

# A   Data attribution methods

In Section 2.1, we introduce the data attribution methods we implemented in `dattri`. Here, we provide a more detailed introduction to these data attribution methods.

**Influence function.**   First proposed by Koh and Liang [22], the influence function (IF) is defined as:

$$\tau_{\text{IF}}(x, \mathcal{S}; f_\Theta) = \left[ g_{f_\Theta}(x_j)^\top H_{f_\Theta}^{-1} g_{f_\Theta}(x) : x_j \in \mathcal{S} \right],$$

where $g_{f_\Theta}(x)$ is the vector gradient of loss function with respect to the parameters $\Theta$ evaluated at testing sample $x$, $H_{f_\Theta}^{-1}$ is the inverse hessian matrix with respect to the training set, and $g_{f_\Theta}(x_j)^\top$ is the vector gradient evaluated at training sample $x_j$. The product of the first two terms is an inverse-hessian-vector-product (IHVP) problem, where we implement the explicit calculation, conjugate gradients, LiSSA method, and Arnoldi iteration approximation to solve it. We refer readers to check the original papers [22, 30] for more details.

**TracInCP.**   Pruthi et al. [29] proposes TracIn, which aims at tracing the training samples' influence throughout the process of model training. Multiple checkpoints (i.e., $\{\Theta^{(i)}\}_{i=1}^I$) stored during the training process are combined to give the final estimate of training data attribution. The formulation can be summarized as follows:

$$\tau_{\text{TracInCP}}(x, \mathcal{S}; \{f_{\Theta^{(i)}}\}_{i=1}^I) = \left[ \sum_{i=1}^I \eta_i \cdot g_{f_{\Theta^{(i)}}}(x)^\top g_{f_{\Theta^{(i)}}}(x_j) : x_j \in \mathcal{S} \right],$$

where $\eta_i$ is the step size used by the optimizer during the $(i-1)$-th and $i$-th checkpoints, $g_{f_{\Theta^{(i)}}}(x)^\top$ is the vector gradient of model output $f_{\Theta^{(i)}}$ with respect to the parameters $\Theta^{(i)}$ evaluated at testing sample $x$, and $g_{f_{\Theta^{(i)}}}(x_j)^\top$ is the gradient of loss function with respect to the $i$-th checkpoint's parameters evaluated at training sample $x_j$.

**Grad-Dot.**   Grad-Dot is proposed by Charpiat et al. [5], which can be seen as a special case of TracIn where only the final checkpoint of the trained model is used to compute the data attribution score. Formally, it can be fomulatd as follows:

$$\tau_{\text{Grad-Dot}}(x, \mathcal{S}; f_\Theta) = \left[ g_{f_\Theta}(x)^\top g_{f_\Theta}(x_j) : x_j \in \mathcal{S} \right],$$

where $g_{f_\Theta}(x)$ is the vector gradient of model output $f_\Theta$ with respect to the parameters $\Theta$ evaluated at testing sample $x$, and $g_{f_\Theta}(x_j)^\top$ is the gradient of loss function evaluated at the training sample $x_j$.

**Grad-Cos.**   Similar to Grad-Dot [5] but with a normalization operation performed on the gradients, Grad-Cos can be summarized as follows:

$$\tau_{\text{Grad-Cos}}(x, \mathcal{S}; f_\Theta) = \left[ \frac{g_{f_\Theta}(x)^\top g_{f_\Theta}(x_j)}{\|g_{f_\Theta}(x)\| \|g_{f_\Theta}(x_j)\|} : x_j \in \mathcal{S} \right],$$

where $g_{f_\Theta}(x)$ is the vector gradient of model output $f_\Theta$ with respect to the parameters $\Theta$ evaluated at testing sample $x$, and $g_{f_\Theta}(x_j)^\top$ is the vector gradient of loss function evaluated at the training sample $x_j$.

It is worth pointing out that Grad-Cos is theoretically no better than a random guess baseline on the noisy label detection task. Specifically, noisy label detection by data attribution methods relies on the *self-influence score*, i.e., the influence/attribution score of a training data point with the target data point setting as itself. As a result, the gradient of the training data point is the same as that of the target data point, which leads to a Grad-Cos score always being 1. Therefore, Grad-Cos score has no discrimination power for the noisy label detection task.

**Representer point selection (RPS-L2).**   Yeh et al. [36] proposes representer point selection (RPS), which leverages the representer theorem to decompose the pre-activation prediction layer's output in a neural network model by a linear combination over kernel evaluations at the training data points. Formally, we can represent the model output function $f_\Theta$ as $f_{\{\Theta_1, \Theta_2\}} = \Phi(x_j, \Theta_1) := \Theta_1 h_j$ with

$\Theta_1$ as the parameters of the last linear classification layer and $h_j$ as the last intermediate layer feature for input $x_j \in \mathcal{S}$. Note that here $h_j = \Phi_2(x_j, \Theta_2)$ and $\Theta_2$ are all the parameters to generate the last intermediate layer from the input $x_j$. The authors stated that if $\Theta^*$ is a stationary point of the following L2-constrained optimization problem:

$$\min_{\Theta} \left\{ \frac{1}{n} \sum_{j=1}^{n} \mathcal{L}(x_j, \Theta) + \lambda \|\|\Theta_1\|_2^2 \right\},$$

for a pre-specified loss function $\mathcal{L}(\cdot)$ and a regularization coefficient $\lambda > 0$, then the representer values, which can be considered as the data attribution scores, can be computed as follows:

$$\tau_{\text{RPS-L2}}(x, \mathcal{S}; f_{\Theta}) = \left[ \frac{1}{-2\lambda n} \frac{\partial \mathcal{L}(x, \Theta^*)}{\partial \Phi(x_j, \Theta^*)} \cdot h_j^T h : x_j \in \mathcal{S} \right],$$

where $n$ is the number of training points in training set $\mathcal{S}$, $h_j$ is the last intermediate layer feature for training sample $x_j$ and $h$ is the last intermediate layer feature for testing sample $x$.

**Tracing with the Randomly-projected After Kernel (TRAK).** This is a state-of-the-art data attribution method introduced by Park et al. [27]. Formally, TRAK is formulated as follows:

$$\tau_{\text{TRAK}}(x, \mathcal{S}; \{f_{\Theta^{(i)}}\}_{i=1}^{I}) = \left( \frac{1}{I} \sum_{i=1}^{I} \mathbf{Q}_{f_{\Theta^{(i)}}} \right) \left( \frac{1}{I} \sum_{i=1}^{I} \phi_{f_{\Theta^{(i)}}} \left( \Phi_{f_{\Theta^{(i)}}}^{\top} \Phi_{f_{\Theta^{(i)}}} \right)^{-1} \Phi_{f_{\Theta^{(i)}}}^{\top} \right),$$

where $\Theta^{(i)}$ are parameters of models independently trained on the training set $\mathcal{S}$; $\mathbf{Q}_{f_{\Theta^{(i)}}}$ is a diagonal matrix with each diagonal element corresponding to the "one minus correct-class probability" of a training data point under model $f_{\Theta^{(i)}}$; $\phi_{f_{\Theta^{(i)}}}$ is the vector gradient of $f_{\Theta^{(i)}}(x)$ with respect to $\Theta^{(i)}$; and $\Phi_{f_{\Theta^{(i)}}}$ is the matrix of the vector gradients of $f_{\Theta^{(i)}}(x_j)$ stacked over the training samples $x_j \in S$.[4]

## B  Details on linear datamodeling score (LDS)

Introduced by Park et al. [27], the *linear datamodeling score* (LDS) is proposed to probe the data attribution method's ability to make counterfactual predictions based on the attribution score derived from the learned model output function $f_{\Theta}$ and the corresponding dataset to train $f_{\Theta}$. Formally, the *attribution-based output predictions* of the model output function $f_{\Theta_{\mathcal{S}'}}$ is defined as follows:

$$g_{\tau}(x, \mathcal{S}'; \mathcal{S}) \triangleq \sum_{i:x_i \in \mathcal{S}'} \tau(x, \mathcal{S}; f_{\Theta})_i, \tag{1}$$

where $\mathcal{S}$ is the training set, $\mathcal{S}' \subseteq \mathcal{S}$ is a subset of $\mathcal{S}$ and $f_{\Theta_{\mathcal{S}'}}$ is the model output function with $\Theta_{\mathcal{S}'}$ learned from $\mathcal{S}'$. Intuitively, $g_{\tau}(x, \mathcal{S}'; \mathcal{S})$ computes the overall attribution of the subset $\mathcal{S}'$ on example $x$ by summing up the individual attribution scores for training samples in the set $\mathcal{S}'$, which can be seen as a powerful indicator of the model prediction on $x$ (i.e., $f_{\Theta_{\mathcal{S}'}}(x)$) if the data attribution method performs properly. The *linear datamodeling score* (LDS) is defined to measure the predictive power of $g_{\tau}(x, \mathcal{S}'; \mathcal{S})$ and can be formalized as follows:

**Definition B.1** (Linear datamodeling score). *Given a training set $\mathcal{S}$, a model output function $f_{\Theta}$, and a corresponding data attribution method $\tau$. Let $\{\mathcal{S}_1, \ldots, \mathcal{S}_m : \mathcal{S}_j \subseteq \mathcal{S}\}$ be $m$ randomly sampled subsets of $\mathcal{S}$, each of size $\alpha \times n$ for some fixed $\alpha \in (0, 1)$. The linear datamodeling score (LDS) of $\tau$ for a specific example $x$ is defined as*

$$\text{LDS}(\tau, x) \triangleq \rho(\{f_{\Theta_{\mathcal{S}_j}}(x) : j \in [m]\}, \{g_{\tau}(x, \mathcal{S}_j; \mathcal{S}) : j \in [m]\}),$$

*where $\rho$ is the Spearman rank correlation [32], $f_{\Theta_{\mathcal{S}_j}}$ is the model output function with $\Theta_{\mathcal{S}_j}$ learned from $\mathcal{S}_j$ and $g_{\tau}(x, \mathcal{S}_j; \mathcal{S})$ is defined in Equation (1).*

For all benchmark experiments that are evaluated by LDS, we use 50 models that are independently trained on random subsets with size half of the full dataset (i.e., we set $m = 50$ and $\alpha = 0.5$ in Theorem B.1).

---

[4]Some details of the TRAK formulation are omitted here. Please refer to the original paper for more details.

# C  Detailed benchmark experiment setup

In this section, we provide the detailed setup for each benchmark experiment mentioned in Section 3.3. The experiment is run on an internal server with an Nvidia A40 GPU for around 500 hours.

**LR/MLP on MNIST-10.**  Firstly, we use a simple logistic regression architecture, which consists of a linear layer and a sigmoid activation layer. The linear layer has 7840 parameters in total. We employ an SGD optimizer with a learning rate of 0.01, momentum 0.9 and batch size 64 to train the model for 20 epochs. Secondly, we use a 3-layer MLP with hidden layer sizes equal to 128 and 64 and placed dropout layers after the first two linear layers with a rate of 0.1, which results in a total of about 0.11M parameters. We employ the same optimizer and batch size to train this MLP classifier for 50 epochs.

**ResNet-9 on CIFAR-2/CIFAR-10.**  For CIFAR-2 experiment, we construct the CIFAR-2 dataset by sampling from the CIFAR-10 dataset that only include the "cat" and "dog" classes. We train a Resnet-9 model [13] on this dataset, which has a total of about 0.38M parameters. We place dropout layers after all convolution layers with a rate of 0.1. We employ an SGD optimizer with a learning rate of 0.01, momentum of 0.9, and batch size 64 to train this MLP classifier for 50 epochs. The model has roughly 4.83M trainable parameters, and we train this model for 50 epochs. For the CIFAR-10 experiment, we follow the setting without any subsampling on the dataset and change only the last linear layer of the ResNet-9 model for 10-way classifications.

**Music Transformer on MAESTRO.**  For the MAESTRO experiment, we utilize the MIDI and Audio Edited for Synchronous TRacks and Organization (MAESTRO) dataset (v2.0.0) [12] and construct a Music Transformer corresponding to the default setting specified in [14]. Specifically, the number of transformer layers equals 6, the number of multi-heads equals 8, the input feature size is 512, and the dimension of the feedforward network is 1024. For data pre-processing, we follow the experiment setup detailed in [6]. To be more specific, we define a vocabulary set of size equal to 388, which includes "NOTE ON" and "NOTE OFF" events for 128 different pitches, 100 "TIME SHIFT" events, and 32 "VELOCITY" events. The raw data is processed as sequences of about 90K events. When training the Music Transformer, the batch size is set to be 64, and the model is trained by a classic seq2seq loss function. We employ an Adam optimizer with a learning rate equal to 1e-4, $\beta_1 = 0.9$ and $\beta_2 = 0.98$. We apply zero warm-up steps, and we train the model for 20 epochs. For music event generation, we use 178 samples from the official testing dataset as prompts to generate music with a single event.

**NanoGPT on Shakespeare.**  For the Shakespeare experiment, we utilize the Tiny Shakespeare dataset [20] and build a nanoGPT model [21]. Specifically, the number of transformer layers is equal to 6, and the number of multi-heads is equal to 6. For data pre-processing, we define the block size to be 256 and the full dataset can be processed into 3921 samples. During the training, the batch size is set to 32, and we employ an Adam optimizer with a learning rate equal to 1e-3, $\beta_1 = 0.9$ and $\beta_2 = 0.99$. We train the model for 5000 samples. The training follows the default setting of `https://github.com/karpathy/nanoGPT`.

For data licenses, the MNIST-10 dataset holds a CC BY-SA 3.0 license, the CIFAR-10 dataset holds a CC-BY 4.0 license, the MAESTRO dataset holds a CC BY-NC-SA 4.0 license, and the Shakespeare dataset is in public domain.

## C.1  Data attribution methods hyperparameters

The experimental results of each data attribution method record the best performance over a hyperparameter search space. Here, we list the hyperparameter search space.

- "explicit": search "regularization" among [1e-1, 1e-2, 5e-3, 1e-3, 1e-4, 1e-5]
- "CG": search "regularization" among [1e-1, 1e-2, 5e-3, 1e-3, 1e-4, 1e-5], "max_iter": 10.
- "LiSSA": search among [{"recursion_depth": 500, "batch_size": 10}, {"recursion_depth": 500, "batch_size": 50}]
- "Arnoldi": search "regularization" among [1e-1, 1e-2, 5e-3, 1e-3, 1e-4, 1e-5], "max_iter": 50.

- "RPS-L2": search "L2 regularization strength": among [10, 1, 1e-1, 1e-2, 1e-3, 1e-4], "feature normalization": among [True, False].

- "TRAK": search "projection dimension" among [512, 2048].

# D   Runtime and memory

We report the runtime and memory usage of different data attribution methods for MLP on MNIST-10 and ResNet-9 on CIFAR-2. For the IF family, Arnoldi is faster and can be scaled to larger settings than other variants in the IF family, such as CG and Explicit. The TracIN family is mostly faster and requires less memory than IF. The RPS family is the fastest and lightest because it only cares about the last layer parameters. The runtime of TRAK increases nearly linearly with respect to the number of ensemble models. In terms of memory, TRAK has a constant memory overhead due to the gradient projection, which grows with the model size but does not grow with the number of ensemble models, and another part of memory cost that grows linearly with the number of ensemble models.

Table 4: Runtime and Memory of different data attribution methods on MNIST-10+MLP and CIFAR-2+ResNet-9 experiments. The numbers are recorded on a single A40 GPU with 48GB memory.

| Family | Algorithms | MNIST-10+MLP | | CIFAR-2+ResNet-9 | |
|---|---|---|---|---|---|
| | | Runtime | Peak Memory | Runtime | Peak Memory |
| IF | Explicit | X | OOM | X | OOM |
| | Conjugate Gradients (CG) | 610.79s | 966.99M | X | OOM |
| | LiSSA | 498.53s | 323.54M | X | OOM |
| | Arnoldi | 455.80s | 250.50M | 732.92s | 16405.30M |
| TracIn | TracInCP-10 | 85.89s | 214.24M | 2250.02s | 9475.81M |
| | Grad-Dot | 9.06s | 214.24M | 226.50s | 9476.98M |
| | Grad-Cos | 8.76s | 214.24M | 235.01s | 9475.54M |
| RPS | RPS-L2 | 1.08s | 72.09M | 1.22s | 296.81M |
| TRAK | TRAK-1 | 1.83s | 206.99M | 36.98s | 7048.40M |
| | TRAK-10 | 13.71s | 598.67M | 316.22s | 7440.87M |
| | TRAK-50 | 67.74s | 2236.08M | 1549.24s | 9077.21M |

# E   Demo of using low-level utility functions to build new data attribution methods

We demonstrate how developers can build new data attribution methods using low-level utility functions implemented in `dattri`. In this example, we will replace the random projection used in TRAK with the Arnoldi projection, leading to a new data attribution method.

```
grad_t = self.grad_loss_func(parameters, train_batch_data)
grad_t = torch.nan_to_num(grad_t) / self.norm_scaler
grad_p = random_project(
grad_t,
train_batch_data[0].shape[0],
**self.projector_kwargs,
)(grad_t, ensemble_id=ckpt_seed).clone().detach()
```

Demo 3: The original TRAK using `random_project` as the projection method.

```
        grad_t = self.grad_loss_func(parameters, train_batch_data)
        grad_t = torch.nan_to_num(grad_t) / self.norm_scaler
        grad_p = arnoldi_project(
        grad_t,
        self.target_func,
        parameters,
        **self.projector_kwargs,
        )(grad_t).clone().detach()
```

Demo 4: A new variant of TRAK using `arnoldi_project` as the projection method.

# F    Quick start demo

The following is a quick start demo using `dattri` to conduct data attribution for the logistic regression model on MNIST.

```python
import torch
from torch import nn

from dattri.algorithm import IFAttributorCG
from dattri.task import AttributionTask
from dattri.benchmark.datasets.mnist import train_mnist_lr, create_mnist_dataset
from dattri.benchmark.utils import SubsetSampler

dataset_train, dataset_test = create_mnist_dataset("./data")

train_loader = torch.utils.data.DataLoader(
    dataset_train,
    batch_size=1000,
    sampler=SubsetSampler(range(1000)),
)
test_loader = torch.utils.data.DataLoader(
    dataset_test,
    batch_size=100,
    sampler=SubsetSampler(range(100)),
)

model = train_mnist_lr(train_loader)
```

```python
 1  def f(params, data_target_pair):
 2      x, y = data_target_pair
 3      loss = nn.CrossEntropyLoss()
 4      yhat = torch.func.functional_call(model, params, x)
 5      return loss(yhat, y)
 6
 7  task = AttributionTask(loss_func=f,
 8                         model=model,
 9                         checkpoints=model.state_dict())
10
11  attributor = IFAttributorCG(
12      task=task,
13      max_iter=10,
14      regularization=1e-2
15  )
16
17  attributor.cache(train_loader)
18  score = attributor.attribute(train_loader, test_loader)
```

Demo 5: Quick start demo. The yellow code block is where we apply `dattri` for data attribution.

# G Uncertainty analysis

We report the uncertainty of LDS scores for different data attribution methods under two sources of randomness, *algorithm randomness* and *ground-truth randomness*. For algorithm randomness, we train independent models with five different random seeds as the models to be attributed and evaluate the data attribution methods on the same sets of randomly sampled subsets. For ground-truth randomness, we fix the seed for model training but evaluate the data attribution methods on five independent sets of randomly sampled subsets.

As shown in Figure 5, we report the error bars of the LDS scores on two experimental settings, MLP on MNIST-10 and RseNet-9 on CIFAR-2, coupled with the aforementioned two sources of randomness. Overall, the error bars are small among all settings, except for those methods that have very poor performance (with LDS scores close to 0).

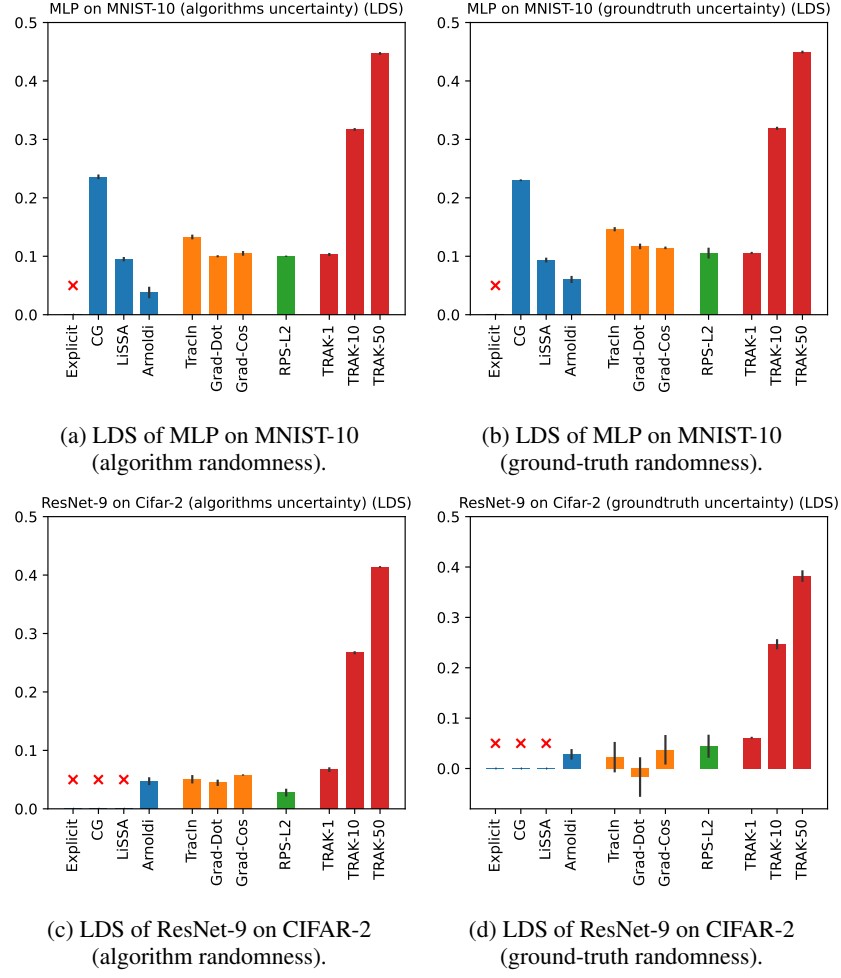

(a) LDS of MLP on MNIST-10 (algorithm randomness).

(b) LDS of MLP on MNIST-10 (ground-truth randomness).

(c) LDS of ResNet-9 on CIFAR-2 (algorithm randomness).

(d) LDS of ResNet-9 on CIFAR-2 (ground-truth randomness).

Figure 5: The LDS evaluation of different data attribution methods on MLP trained on MNIST-10 and ResNet-9 trained on CIFAR-2 with two sources of randomness. The error bars reflect the standard error of the mean.

