# OpenReview forum: "$\texttt{dattri}$: A Library for Efficient Data Attribution"
_NeurIPS.cc/2024/Datasets_and_Benchmarks_Track — NeurIPS 2024 Track Datasets and Benchmarks Spotlight_

### Official Review · Reviewer_W7KE · 2024-07-15
**A Comprehensive Open-source Library for Gradient-based Data Attribution**

**Rating:** 7
**Confidence:** 4
**Correctness:** Yes
**Clarity:** The writing is clear and easy to follow.

**Review:**

The dattri library provides implementation of various data attribution methods and includes a comprehensive benchmark suite that supports a wide range of machine learning tasks and robust metrics such as LOO correlation, LDS, and AUC, facilitating a thorough evaluation across diverse contexts. It also provides easy-to-use functionality such as Hessian-vector products, and TRAK computation.

However, the focus on computationally efficient methods excludes potentially valuable game-theoretic approaches like Data Shapley and Data Banzhaf. Additionally, the paper could benefit from more detailed validation through extended error analysis to better establish the methods' reliability and stability.

**Strengths:**

* Unified API and Ease of Use: dattri simplifies the integration of various data attribution methods into existing machine learning workflows in pytorch. The library modularizes utility functions like Hessian-vector products and TRAK, which is a significant advancement in user experience for developers and researchers. They also provide model checkpoints.

* Comprehensive Benchmarking: The inclusion of diverse machine learning tasks (e.g., image classification, text, and music generation) and comprehensive evaluation metrics (e.g., LOO correlation, LDS, AUC for noisy label detection) in the benchmark suite is commendable, allowing for a thorough assessment of attribution methods across varied contexts.

**Additional Feedback:**

None

**Documentation:**

Yes

**Limitations:**

Yes

**Opportunities For Improvement:**

* Including Retraining-based Methods: While the focus on computationally efficient methods is practical, However, the inclusion of game-theoretic, retraining-based methods such as Data Shapley, Data Banzhaf, and Datamodel would significantly enrich the library's capabilities.

* Additional Evaluation Metrics: Incorporating counterfactual analysis could enhance the library's evaluation capabilities. Removing data points based on their attribution scores, e.g. the top 25% of data, and then retraining the model, would provide deeper insights into the true impact of specific data points on model performance.

* Extended Validation and Error Analysis: The paper lacks error bars and variance. This is crucial for neural networks when validating the stability and reliability of the attribution methods under different conditions.

**Relation To Prior Work:**

Yes. It is addressed in section 2.2.

**Summary And Contributions:**

This paper introduces a public library, **dattri**, an open-source library designed to streamline the development, benchmarking, and deployment of data attribution methods. It offers a unified API for PyTorch users, modularizes common utility functions, and provides a comprehensive benchmark suite with pre-trained models. The data attribution method mostly includes gradient-based approaches, e.g. influence function(IF), TRAK, and TracIn. The performance metrics include LOO, AUC, and LDS.

---

> ### Author Rebuttal · Authors · 2024-08-17
>
> We thank the reviewer for taking time to review our paper and providing thoughtful feedback. We address your individual comments below.
>
> > Improvement 1: Game-theoretic and retraining-based methods
>
> While we acknowledge that game-theoretic and retraining-based methods include some seminal data attribution methods, most of them are computationally demanding due to the requirement for a large number of model retrainings. From the benchmarking perspective, the high computational cost means that they cannot be benchmarked on most of the experiment settings included in our library, which is specifically tailored for evaluating efficient data attribution methods.
>
> In addition, there already exists OpenDataVal [1], a benchmark suite that focuses on game-theoretic methods. There, the benchmark settings are mostly shallow classifiers on low-dimensional tabular data or condensed vector embeddings of text/image. OpenDataVal implements relatively few gradient-based methods, making OpenDataVal and our library serve complementary purposes.
>
> With that said, one game-theoretic method, KNN Shapley [2], stands out for its computational efficiency as it avoids the need for model retraining. We have therefore included KNN Shapley in our latest library (https://github.com/TRAIS-Lab/dattri/blob/main/dattri/algorithm/data_shapley.py). We have also done some preliminary experiments for KNN Shapley. Our implementation achieves an AUC of 0.991 for the noisy label detection task on MNIST.
>
>
> > Improvement 2: Additional evaluation metrics
>
> Thanks for the suggestion. We realize that the metric suggested by the reviewer is close to a metric (named *brittleness*) being used in recent literature [3, 4], which measures the minimal number of influential training data point removal that can flip the model prediction. We have added this metric in our latest codebase (https://github.com/TRAIS-Lab/dattri/blob/main/dattri/metrics/britteness.py) but we haven’t got a chance to conduct a comprehensive experiment with this metric due to the high computation cost associated with it – it requires a unique retraining after each removal of top influential data points for each data attribution method. We hope to follow up with more results later.
>
> > Improvement 3: Extended validation and error analysis
>
> Following the reviewer’s suggestion, we have conducted additional experiments on some experimental settings for error analysis. Overall, we find the variance of the experimental results is small. Please see the accompanying PDF for the results.
>
> > References
>
> [1] Jiang, et al. "OpenDataVal: a unified benchmark for data valuation." NeurIPS 2023.
>
> [2] Jia, et al. "Efficient task-specific data valuation for nearest neighbor algorithms." VLDB 2019.
>
> [3] Ilyas, et al. "Datamodels: Predicting predictions from training data." ICML 2022.
>
> [4] Choe, et al. "What is Your Data Worth to GPT? LLM-Scale Data Valuation with Influence Functions." Arxiv 2024.

---

> > ### Comment · Reviewer_W7KE · 2024-08-18
> >
> > Thank you for addressing my comments. I have raised my score accordingly.

---

> > > ### Author Response · Authors · 2024-08-18
> > > **Thank you!**
> > >
> > > Thank you for acknowledging our response and updating the assessment!

---

### Official Review · Reviewer_eHpZ · 2024-07-27
**useful toolbox for data attribution methods**

**Rating:** 6
**Confidence:** 2
**Correctness:** ~
**Clarity:** ~

**Review:**

Overall, the paper is well-written. The API is explained properly, and the provided low-level utility functions are a useful addition to the library, helpful for further data attribution research. I have a few questions:

1. As mentioned in Table 2, there are no game theoretic methods in this suite. Why?
2. As mentioned in line 113-114, there is a significant overlap with the influenciae library. Could you please highlight the major differences between this toolbox and influenciae, for eg. in terms of methods implemented?

**Strengths:**

~

**Additional Feedback:**

~

**Documentation:**

~

**Ethics:**

~

**Limitations:**

~

**Opportunities For Improvement:**

Possible typo: line 96-98: repeated "assumes hessian matrix to be an identity matrix"

**Relation To Prior Work:**

~

**Summary And Contributions:**

Data attribution methods aim to quantify the influence of a single training example on the output of the model. The authors propose dattri, an open source library implementing popular data attribution methods and common low-level utilities needed for the design of such methods, along with a benchmark suite evaluating these methods.

---

> ### Author Rebuttal · Authors · 2024-08-17
>
> We thank the reviewer for taking time to review our paper and providing thoughtful feedback. We address your individual comments below.
>
> > Question 1: No game-theoretic methods
>
> While we acknowledge that game-theoretic methods represent an important family of data attribution methods, most of them are computationally demanding due to the requirement for a large number of model retrainings. From the benchmarking perspective, the high computational cost means that they cannot be benchmarked on most of the experiment settings included in our library, which is specifically tailored for evaluating efficient data attribution methods.
>
> In addition, there already exists OpenDataVal [1], a benchmark suite that focuses on game-theoretic methods. There, the benchmark settings are mostly shallow classifiers on low-dimensional tabular data or condensed vector embeddings of text/image. OpenDataVal implements relatively few gradient-based methods, making OpenDataVal and our library serve complementary purposes.
>
> With that said, one game-theoretic method, KNN Shapley [2], stands out for its computational efficiency as it avoids the need for model retraining. We have therefore included KNN Shapley in our latest library (https://github.com/TRAIS-Lab/dattri/blob/main/dattri/algorithm/data_shapley.py). We have also done some preliminary experiments for KNN Shapley. Our implementation achieves an AUC of 0.991 for the noisy label detection task on MNIST.
>
> > Question 2: Comparison with Influenciae
>
> A key distinction between `dattri` and `Influenciae` lies in their design objectives. While `Influenciae` primarily targets practitioners for the application of existing data attribution methods, `dattri` is purposefully designed to meet the needs of the developers and researchers. This broader focus drives many novel design features, such as the inclusion of modular utility functions to facilitate new algorithm development, a more comprehensive benchmarking suite (more metrics, generative tasks) for better performance evaluations, and the provision of retrained model checkpoints to minimize benchmarking costs. In addition, `Influenciae` is implemented in Tensorflow while `dattri` is implemented in PyTorch, which is arguably more popular than Tensorflow for machine learning research in recent years.
>
> > Improvement 1: Typo
>
> Thanks for pointing it out. We will fix it in our updated draft.
>
> > References
>
> [1] Jiang, et al. "OpenDataVal: a unified benchmark for data valuation." NeurIPS 2023.
>
> [2] Jia, et al. "Efficient task-specific data valuation for nearest neighbor algorithms." VLDB 2019.

---

> > ### Comment · Reviewer_eHpZ · 2024-08-18
> >
> > Thank you for your response. I will keep my score the same for now since I still feel that there is significant overlap with prior work (influenciae library) in terms of implemented methods.

---

> > > ### Author Response · Authors · 2024-08-26
> > > **Please let us know if our follow-up response has addressed your concern**
> > >
> > > Dear Reviewer eHpZ,
> > >
> > > Thank you for your valuable review and follow-up comment regarding the comparison with `Influenciae`. We have carefully addressed this in our follow-up response, highlighting that our library offers new SOTA methods, **significantly more comprehensive benchmark datasets, tasks, and metrics**, as well as other novel features (details are provided in our follow-up response). We would be grateful if you could kindly review our follow-up and let us know if you have any further concerns. Thank you for your time and consideration!

---

> ### Author Response · Authors · 2024-08-18
> **Follow-up response**
>
> Thank you for the comment.
>
> We do have included more recent state-of-the-art methods such as TRAK and DataInf compared to Influenciae. The newly added KNN Shapley is not included in Influenciae either.
>
> With that said, we believe the more salient distinction between dattri and Influenciae lies in the developer/researcher-oriented design features, including the **significantly more comprehensive** benchmark suite (Influenciae only has one benchmark on noisy label detection on a single dataset CIFAR 10) and the modular utility functions. In the dataset and benchmark track, sometimes a comprehensive and well-designed benchmark suite alone is considered as a significant contribution.
>
> Moreover, our experimental results actually suggest that the noisy label detection task (the sole benchmark adopted by Influenciae) has less discrimination power among different methods in comparison to the LOO and LDS metrics developed in recent literature. Making the LOO and LDS benchmarks efficient in practice is also non-trivial, as a large volume of model retraining is involved. We make the benchmark efficient by providing retraining checkpoints, which is a novel design.
>
> In addition, the choice of PyTorch vs Tensorflow also has a great impact in practice. While this difference may not matter much for a methodology contribution, it is a significant consideration for a benchmark contribution.
>
> We respect the reviewer's opinion while we would like to bring up the above factors to the reviewer's attention, in case those were not fully leveraged into the reviewer's assessment of our work.
>
> We thank the reviewer again for the review and feedback.

---

### Official Review · Reviewer_Ffeh · 2024-08-03
**review for "dattri: A Library for Efficient Data Attribution"**

**Rating:** 7
**Confidence:** 3

**Review:**

the authors address an important need in the rapidly growing field of data attribution. their library provides a valuable resource for both researchers developing new methods and practitioners applying existing ones.

**Strengths:**

1. the unified api design (section 3.1) allows for easy integration of different methods with minimal code changes
2. modularization of common utility functions (section 3.2) facilitates development of new methods
3. comprehensive benchmark suite (section 3.3) covers diverse settings, including generative ai tasks

**Additional Feedback:**

1. consider adding a "quick start" section in the paper, demonstrating how to use dattri for a simple use case
2. discuss potential integration with other popular ml libraries beyond pytorch
3. elaborate on how dattri could be extended to support data attribution for other ml paradigms (e.g., reinforcement learning, unsupervised learning)

**Clarity:**

the paper is well-structured and clearly written. the api usage example in demo 1 is particularly helpful. some technical terms (e.g., "ihvp") could benefit from brief explanations for readers less familiar with data attribution methods.

**Correctness:**

the methods appear to be correctly implemented, and claims are supported by experimental results. however, tbf a full correctness review of the library is beyond the scope of a conference submission.

**Documentation:**

the description of the library's components and benchmark suite is thorough. the authors mention that code is available on github, which is crucial for reproducibility.

**Limitations:**

the authors acknowledge limitations in section 5, particularly the current coverage of methods and benchmark settings. they could further discuss potential biases in their benchmark suite.

**Opportunities For Improvement:**

1. why exclude game-theoretic methods like data shapley? could these be included as a separate module for completeness?
1. in figure 2, why does the performance of all methods drop so significantly from lr to mlp? is this expected?
1. for the larger models in figure 3, only trak shows non-trivial performance. could you discuss potential reasons for this?
1. how does the computational efficiency of different methods compare? a runtime comparison would be valuable.
1. in section 3.1, how does the @flatten_func decorator work? could you provide more details on its implementation?
1. for the benchmark suite (section 3.3), how were the specific datasets and models chosen? are they representative of common ml tasks?
1. in figure 4, why does grad-cos perform so poorly compared to other methods?
1. include a table comparing runtime and memory usage of different methods across benchmark settings
1. provide code snippets demonstrating how to implement a new data attribution method using dattri's utility functions
1. discuss potential limitations of the current evaluation metrics and suggest directions for developing new ones
1. compare your findings to those in "a survey on data selection for language models" by albalak et al. (https://arxiv.org/abs/2402.16827) regarding the impact of data on model performance

**Relation To Prior Work:**

the authors provide a comprehensive comparison to existing libraries in table 2. they could further discuss how their benchmark results compare to those reported in original papers for each method.

**Summary And Contributions:**

the authors present dattri, an open-source library for data attribution methods in machine learning. the main contributions are:

1. a unified and user-friendly api for integrating data attribution methods into pytorch-based ml pipelines
2. modularized implementations of low-level utility functions common in data attribution methods
3. a comprehensive benchmark suite with diverse evaluation metrics and experimental settings

---

> ### Author Rebuttal · Authors · 2024-08-17
>
> We thank the reviewer for taking time to review our paper and providing thoughtful feedback. We address your individual comments below.
>
> > Improvement 1: Game-theoretic methods
>
> While we acknowledge that game-theoretic methods represent an important family of data attribution methods, most of them are computationally demanding due to the requirement for a large number of model retrainings. From the benchmarking perspective, the high computational cost means that they cannot be benchmarked on most of the experiment settings included in our library, which is specifically tailored for evaluating efficient data attribution methods.
>
> In addition, there already exists OpenDataVal [1], a benchmark suite that focuses on game-theoretic methods. There, the benchmark settings are mostly shallow classifiers on low-dimensional tabular data or condensed vector embeddings of text/image. OpenDataVal implements relatively few gradient-based methods, making OpenDataVal and our library serve complementary purposes.
>
> With that said, one game-theoretic method, KNN Shapley [2], stands out for its computational efficiency as it avoids the need for model retraining. We have therefore included KNN Shapley in our latest library (https://github.com/TRAIS-Lab/dattri/blob/main/dattri/algorithm/data_shapley.py). We have also done some preliminary experiments for KNN Shapley. Our implementation achieves an AUC of 0.991 for the noisy label detection task on MNIST.
>
> > Improvement 2: Performance on LR and MLP in Figure 2
>
> Most of the gradient-based data attribution methods rely on the approximations by linearizing the model (to various extent). The approximations are better for Logistic Regression (LR) than to more complex models like Multilayer Perceptron (MLP), which is likely the reason for the performance drop on MLP compared to LR.
>
> Some previous work also verified this performance degradation. One of the most direct results of LOO is Figure 1 in [3]. For LDS, the results in our draft are also comparable as the results in [4].
>
> > Improvement 3: Discussion about the performance gap between TRAK and others in Figure 3.
>
> Experiment settings in Figure 3 include larger models and more complex datasets, which is more challenging than the ones in Figure 2.
>
> Many variants of influence function suffer from OOM issues on larger models. Lightweight methods like TracIN, Grad-Cos, and Grad-Dot essentially assume the hessian matrix to be an identity matrix, which can be efficient while sacrificing their effectiveness, especially for larger and more complex models. TRAK [4] is a state-of-the-art method among gradient-based data attribution methods and features a good efficacy-efficiency trade-off. The results in our draft are comparable as the results in [4].
>
> > Improvement 4 and 8: Runtime and memory of different methods
>
> We have included additional experiments to record the runtime and memory of different methods. Please see the details in the Section D of the PDF accompanying this rebuttal.
>
> > Improvement 5: Elaborate flatten_func
>
> In `dattri` implementation, we heavily utilize `torch.func`, a composable function transform on PyTorch. It features many advantages over the autograd mechanism in PyTorch, such as per-sample-gradients. However, it is more convenient to use `torch.func` on a transformation of standard PyTorch models. flatten_func is a decorator that facilitates this transformation.
>
> Specifically, flatten_func is a parameterized python decorator. It stores the model parameters structure and transforms the flattened parameters input to a model parameter dictionary that can be directly used in `torch.func.functional_call`. This is done by using the functools.wrap to modify users’ functions.
>
> > Improvement 6: Choice of datasets and models
>
> Classification tasks based on LR, MLP, and ResNet models on MNIST-10 and CIFAR-2/10 datasets have been commonly used in the data attribution literature [3, 4, 5]. For the generation tasks, there are relatively few data attribution studies focusing on generation tasks yet so there hasn’t been a standard setting of dataset + model. We include several models and datasets in text and music generation [6] to increase the variety of generation tasks. We also intentionally include some smaller-scale settings for the purpose of facilitating fast iterations of model development and evaluation.
>
> > Improvement 7: Poor performance of Grad-Cos in Figure 4
>
> Thanks for the question. With some further investigation, we find that Grad-Cos is theoretically no better than a random guess baseline on the noisy label detection task.
>
> Specifically, noisy label detection by data attribution methods relies on the **self-influence score**, i.e., the influence/attribution score of a training data point with the target data point setting as itself. As a result, the gradient of the training data point is the same as that of the target data point, which leads to a Grad-Cos score always being 1. Therefore, Grad-Cos score has no discrimination power for the noisy label detection task.
>
> Our empirical result in Figure 4 aligns with the analysis above, where the AUC of Grad-Cos is close to 0.5 (random guess) for all experiment settings.
>
> > References
>
> [1] Jiang, et al. "OpenDataVal: a unified benchmark for data valuation." NeurIPS 2023.
>
> [2] Jia, et al. "Efficient task-specific data valuation for nearest neighbor algorithms." VLDB 2019.
>
> [3] Bae, et al. "If influence functions are the answer, then what is the question?." NeurIPS 2022.
>
> [4] Park, et al. "Trak: Attributing model behavior at scale." ICML 2023.
>
> [5] Samyadeep, et al. "Influence functions in deep learning are fragile." ICLR 2021.
>
> [6] Deng, et al. "Computational copyright: Towards a royalty model for ai music generation platforms." Arxiv 2023.

---

> > ### Author Rebuttal · Authors · 2024-08-17
> >
> > > Improvement 9: A snippet demonstrating the implementation of new data attribution methods using dattri’s utility function
> >
> > Thanks for the suggestion. Please refer to the Section E of the PDF accompanying the rebuttal to see an example where developers can easily incorporate different projection methods in their data attribution method implementation. More specifically, the code snippet shows that the random projection originally used in TRAK can be straightforwardly replaced by the Arnoldi projection, leading to a new variant of data attribution method.
> >
> > > Improvement 10: Limitations of current evaluation metrics and directions of new metrics
> >
> > We first discuss the limitations of the evaluation metrics we currently implemented in `dattri`. The LOO and LDS metrics (especially LOO), while being principled, rely on a large number of retraining models to get the “ground truth”. The AUC metric is tied to a downstream application and is only valid for classification tasks. To alleviate the first limitation, and as one important contribution by `dattri`, we provided retrained model checkpoints associated with the LOO and LDS metrics, so that users can avoid the expensive model retraining process for these evaluation metrics.
> >
> > Regarding directions of new metrics, there is a new metric, named *brittleness*, being used in recent literature [7, 8], which measures the minimal number of influential training data point removal that can flip the model prediction. We have added this metric in our latest codebase (https://github.com/TRAIS-Lab/dattri/blob/main/dattri/metrics/britteness.py). In addition, we also plan to add evaluation metrics related to downstream applications, such as fact tracing [9].
> >
> > > Improvement 11: Comparison to “A survey of data selection for language models”
> >
> > Thanks for pointing us to this survey paper on data selection [10] and we discuss the comparison below. Data selection is an important application of data attribution methods, with empirical success in recent literature [11, 12], which are also mentioned in the survey paper [10]. However, data attribution methods also have other important applications beyond data selection, such as model debugging [13] and data valuation [14]. On the other hand, there are methods beyond data attribution for data selection, such as importance sampling and contrastive data selection [10]. Overall, while there is a certain intersection between this work and the survey paper [8], the scope and focus of the two are drastically different.
> >
> > > Additional Feedback 1: A “quick start” section
> >
> > Thanks for the suggestion. We have added a “quick start” section (Section F in the PDF accompanying this rebuttal) demonstrating how to use `dattri` on a logistic regression model trained on MNIST-10. We will add this section to our updated draft.
> >
> > > Additional Feedback 2: Discuss potential integration with other ML libraries beyond PyTorch
> >
> > Thanks for the suggestion. Many API designs in `dattri` are inspired by `torch.func`, which is a PyTorch version of JAX with an interface that is nearly the same. The modularized low-level utility functions and algorithms can be naturally extended to JAX by a namespace replacement and some minor changes.
> >
> > > Additional Feedback 3: Discuss how dattri could be extended to support data attribution for other ML paradigms (e.g. RL, unsupervised learning)
> >
> > Data attribution for other ML paradigms is a fascinating topic that we are very interested in. However, data attribution research for other ML paradigms has been at a very early stage. From a benchmark perspective, we may need to wait for the methods to be more mature to examine what’s a good unified API for them.
> >
> > > References
> >
> > [1] Jiang, et al. "OpenDataVal: a unified benchmark for data valuation." NeurIPS 2023.
> >
> > [2] Jia, et al. "Efficient task-specific data valuation for nearest neighbor algorithms." VLDB 2019.
> >
> > [3] Bae, et al. "If influence functions are the answer, then what is the question?." NeurIPS 2022.
> >
> > [4] Park, et al. "Trak: Attributing model behavior at scale." ICML 2023.
> >
> > [5] Samyadeep, et al. "Influence functions in deep learning are fragile." ICLR 2021.
> >
> > [6] Deng, et al. "Computational copyright: Towards a royalty model for ai music generation platforms." Arxiv 2023.
> >
> > [7] Ilyas, et al. "Datamodels: Predicting predictions from training data." ICML 2022.
> >
> > [8] Choe, et al. "What is Your Data Worth to GPT? LLM-Scale Data Valuation with Influence Functions." Arxiv 2024.
> >
> > [9] Akyürek, et al. "Towards tracing knowledge in language models back to the training data." Findings of EMNLP 2022.
> >
> > [10] Albalak, et al. "A survey on data selection for language models." TMLR 2024.
> >
> > [11] Xia, et al. "LESS: Selecting influential data for targeted instruction tuning." ICML 2024.
> >
> > [12] Engstrom, et al. "Dsdm: Model-aware dataset selection with datamodels." Arxiv 2024.
> >
> > [13] Koh, et al. "Understanding black-box predictions via influence functions." ICML 2017.
> >
> > [14] Ghorbani, et al. "Data shapley: Equitable valuation of data for machine learning." ICML 2019.

---

> > > ### Author Response · Authors · 2024-08-26
> > > **Please let us know if there is any further question**
> > >
> > > Dear Reviewer Ffeh,
> > >
> > > Thank you for your valuable feedback to our submission! We have carefully addressed your comments in our rebuttal and we would greatly appreciate it if you could review it and consider any further discussion. Thank you for your time and consideration!

---

> > > > ### Comment · Reviewer_Ffeh · 2024-09-01
> > > >
> > > > dear authors thank you for your comments. inclusion of knn shapley as a compromise for game-theoretic methods is a good starting point. your outlined discussion on evaluation metrics and the comparison to the data selection survey would help contextualize dattri's broader scope and potential impact even further. it would be great to see this in a revised manuscript. overall, your responses clarify several concerns and i will raise my score to 7.

---

### Decision · Program_Chairs · 2024-09-26

**Decision:**

Accept (Spotlight)

**Comment:**

This paper proposes. a library that makes it easier to develop data attribution methods, mostly supporting gradient based methods, providing access to some important utilities for Hessian-vector products and TRAK computation. The reviewers all like the paper and championed acceptance, lauding the API design, ease of use, and comprehensive benchmarking. Reviewers were curious about the omission of shapley value based methods and retraining based methods.